# Quality of malaria data in public health facilities in three provinces of Mozambique

**James M. Colborn**[1]*, **Rose Zulliger**[2], **Mariana Da Silva**[3], **Guidion Mathe**[3], **Ana Rita Chico**[1], **Ana Christina Castel-Branco**[1], **Frederico Brito**[4], **Marcel Andela**[1], **Gabriel Ponce de Leon**[5], **Abuchahama Saifodine**[6], **Baltazar Candrinho**[3], **Mateusz M. Plucinski**[5]

1 Clinton Health Access Initiative, Maputo, Mozambique, 2 United States President's Malaria Initiative, United States Centers for Disease Control and Prevention, Maputo, Mozambique, 3 National Malaria Control Program, Ministry of Health, Maputo, Mozambique, 4 UNICEF, Maputo, Mozambique, 5 Malaria Branch and United States President's Malaria Initiative, United States Centers for Disease Control and Prevention, Atlanta, Georgia, United States of America, 6 United States President's Malaria Initiative, United States Agency for International Development, Maputo, Mozambique

* jcolborn.IC@clintonhealthaccess.org

## Abstract

### Background

Malaria data reported through Mozambique's routine health information system are used to guide the implementation of prevention and control activities. Although previous studies have identified issues with the quality of aggregated data reported from public health facilities in the country, no studies have evaluated the quality of routine indicators recorded in health facility registries. This study addresses this issue by comparing indicators calculated from data from exit interviews and re-examinations of patients with data based on registry records from health facilities in order to measure the quality of registry data and data reporting in three provinces in Mozambique.

### Methods

Data were collected from 1,840 outpatients from 117 health facilities in Maputo, Zambézia, and Cabo Delgado Provinces interviewed and examined as part of a malaria-specific health facility survey. Key indicators based on exit interview / re-examination data were compared to the same indicators based on records from health facility registries. Multivariable regression was performed to identify factors associated with indicators matching in re-examination / exit interview data and health facility registries. Aggregated indicators abstracted from facility registries were compared to those reported through the routine health management information system (HMIS) for the same time period.

### Results

Sensitivity of exit interview / re-examination data compared with those recorded in facility registries was low for all indicators in all facilities. The lowest sensitivities were in Maputo, where the sensitivity for recording negative RDT results was 9.7%. The highest sensitivity was for recording positive RDT results in Cabo Delgado, at 75%. Multivariable analysis of

**Data Availability Statement:** The data used in this manuscript are a combination of routine malaria data and survey data on the case management

practices of health care workers in Mozambique. Although these data do not contain personal identifiable information, they are considered to be data under the ownership of the National Malaria Control Program (and thus not public), as their interpretation can be potentially sensitive to the country. Similar data used for similar studies has therefore been considered available upon request, by submitting a letter indicating the proposed use and justification, to the Director of the Mozambique National Malaria Control Program, Dr. Baltazar Candrinho, at candrinhobaltazar@gmail.com.

**Funding:** The authors received no specific funding for this work.

**Competing interests:** The authors have declared that no competing interests exist.

factors associated with agreement between gold standard and registry data showed patients were less likely to be asked about having a fever in the triage ward in Maputo and Cabo Delgado (adjusted Odds Ratio 0.75 and 0.39 respectively), and in the outpatient ward in Cabo Delgado (aOR = 0.37), compared with the emergency department. Patients with positive RDT were also more likely to have RDT results recorded in all three provinces when patients had been managed according to national treatment guidelines during initial examination. Comparison of retrospective data abstracted from facility registries to HMIS data showed discrepancies in all three provinces. The proportion of outpatient cases with suspected and confirmed malaria were similar in registry and HMIS data across all provinces (a relatively low difference between registry and HMIS data of 3% in Maputo and Zambézia), though the total number of all-cause outpatient cases was consistently higher in the HMIS. The largest difference was in Maputo, where a total of 87,992 all-cause outpatient cases were reported in HMIS, compared with a total of 42,431 abstracted from facility registries.

## Conclusion

This study shows that care should be taken in interpreting trends based solely on routine data due to data quality issues, though the discrepancy in all-cause outpatient cases may be indicative that register availability and storage are important factors. As such, simple steps such as providing consistent access and storage of registers that include reporting of patient fever symptoms might improve the quality of routine data recorded at health facilities.

## Introduction

Mozambique's entire population of 29 million people [1] is at risk of malaria. Malaria is a major cause of morbidity and mortality in Mozambique and accounts for 42% of all outpatient consultations, 57% of all paediatric admissions and approximately 23% of all hospital deaths [2]. All-cause, under five mortality in Mozambique has fallen substantially in the recent years, declining from 233/1000 in 1990 to 90/1000 in 2012. Conversely, malaria cases reported through the country's routine health information system have been increasing since 2012, with over 10 million cases reported in 2018. Further complicating the picture is the fact that national malaria prevalence has recently plateaued (40.2% in 2015, 38.9% in 2018 [3,4]), while deaths and inpatient cases resulting from malaria have decreased during this time.

Routine malaria data in Mozambique are initially collected from individual patient data recorded in registers at all public facilities in the country. Key indicators, such as the number of suspect malaria cases and confirmed malaria cases, are then aggregated monthly at each facility on paper forms and sent to District Health authorities, where they are electronically entered into the national routine health information system (HMIS), a DHIS2-based system. HMIS data are accessed by the provincial and National Malaria Control Program (NMCP) to inform decisions on implementation and effectiveness of interventions.

Previous studies in Mozambique have highlighted problems with malaria data quality collected at public health facilities and community health workers, including frequent stockouts of standardized registries, discrepancies between registry and summary aggregate data forms, and lack of consistent definitions for suspected and confirmed cases [5,6]. While most efforts to evaluate data quality have focused on errors with aggregating patient information and

reporting these data into the HMIS [7,8], there are also potential errors in the initial recording of patient-specific information into patient registries. These issues, in addition to the differing trends in routine malaria indicators, highlight the necessity for an evaluation of the quality of malaria case data collected at health facilities to inform analyses and response to routine data in Mozambique.

In April 2018, the NMCP implemented a health facility survey to determine the quality of malaria case management at public health facilities in Maputo, Zambézia, and Cabo Delgado Provinces [9]. In order to provide supplementary data to ongoing DQAs in the country, the NMCP took advantage of the presence of the exit-interview and reexamination data present in the study mentioned above to assess the quality of malaria case data collected in these facilities.

## Methods

### Study design

Data from a previously-described cross-sectional survey of 1,840 patients from 117 public health facilities in Maputo, Zambézia, and Cabo Delgado Provinces in Mozambique [9] were analyzed to assess the quality of registry data and the quality of recording and reporting of patient data specific to malaria case management. The quality of registry data was assessed by comparison of exit-interview data and matched registry entries for the survey participants (Gold Standard), using a previously described methodology [10]; data quality was defined as high sensitivity, specificity, and Cohen's Kappa between the exit-interview and registry data. The quality of reporting was assessed by comparison of summary indicators from one month of retrospective data abstracted from facility monthly registries and corresponding data abstracted from the HMIS for the same one-month period.

### Data collection

As part of the larger study on case management practices, outpatients selected for participation were interviewed after their health facility visit and asked to recall what symptoms they had communicated to the health care worker (HCW). Specifically, they were asked whether or not the HCW had asked about fever or taken their temperature, tests performed, the results of those tests, what they had been prescribed, and how they had been counselled by the HCW to determine whether case management was correct. Patients were also re-examined by a survey clinician who conducted a medical history, measurement of axillary temperature, and testing for malaria with an HRP2-based *Plasmodium falciparum*-specific rapid diagnostic test (RDT; SD Bioline Pf, Yongin, Republic of Korea). These exit interviews (self-report of fever data collected during initial exam) and re-examination data were considered the gold standard for the rest of the analyses done in this study. For each of the patients interviewed during the exit interviews, the corresponding patient data (initial RDT and/or microscopy performed and result) were abstracted from outpatient registries containing all patient's clinical information and laboratory registries, which when available contain RDT and microscopy results. Registry entries were matched based on patient name.

To compare registry data with routine data from the HMIS, the proportion of patients with suspected malaria, the number of patients with malaria, the test positivity rate in suspects with malaria, the number all-cause outpatient consults, and the number of suspected and confirmed malaria cases were abstracted from all available outpatient and laboratory registries encountered by the study team for the month of November 2017 (six months before the month of the survey). This period was selected because it was determined to be the most recent period for which the registers were not still in use by the clinicians, thereby minimizing disruption of clinical activities. Data reported through the HMIS for the same time period were obtained

from the NMCP for each facility included in the study for the same month, and the same set of indicators was calculated in order to compare with those obtained from facility registries.

## Data analysis

To determine how data collected at health facilities compared to the gold standard, data from outpatient re-examinations was cross-tabulated against data from patient interviews and data abstracted from the outpatient and laboratory registries. Seven dichotomous (Yes or No) indicators were created for each patient in the study, according to the following definitions: 1) were patients with fever during re-examination asked about fever during the initial outpatient consultation; 2) were RDT results for the patient present in laboratory registries; 3) did results in laboratory registries match those obtained during re-examination for patients with positive RDT results; 4) for patients with negative RDT results; 5) were microscopy results for the patient present in laboratory registries; 6) did results in laboratory registries match those obtained during re-examination for patients with positive microscopy results and 7) for patients with negative microscopy results.

The sensitivity, specificity, and Cohen's Kappa, which is a measure of agreement between the numbers reported in exit interviews and registries, were calculated considering survey interview data as the gold standard for the seven dichotomous variables described above. For each of these, the sensitivity was defined as the proportion of participants who reported "yes" during the survey interview for a given indicator for whom the same information was recorded in their registry entry. Conversely, specificity was defined as the proportion of participants who reported "no" for a given indicator for whom their registry entry was concordant or, in the case of fever or laboratory test results, absent.

To identify factors associated with matching data in facility registries and exit interview and re-examination data, a multivariable logistic regression was performed on patients with fever during re-examination using three separate dichotomous outcomes: 1) whether patients with fever during re-examination were asked about and had fever recorded during the initial outpatient consultation (a measure of the number of suspected cases recorded in facility registries), 2) whether an RDT result for patients included in the re-examination was recorded in laboratory registries (a measure of the number of test results recorded in facility registries), and 3) amongst patients with positive RDT results, was a positive result recorded in the laboratory registry (a measure of the number of positive test results recorded in facility registries). The following predictor variables were included in the analysis: whether the patient was correctly managed according to national case management guidelines (testing by RDT or microscopy and treatment in accordance with the test result), size of the facility (total monthly consults above or below 1000 patients), age (under 5 years or 5 years and older), the department in which the patient was initially seen (emergency department, outpatient ward, screening/triage ward), and whether or not the patient was classified as having severe malaria (defined as having convulsions as a symptom). Adjusted odds ratios (aORs) with 95% confidence intervals (CI) were calculated for each predictor variable from the results of the logistic regression model.

Values of indicators calculated based on data reported through the HMIS were compared directly to those calculated from data abstracted from facility registries. Differences were calculated as percentages of actual indicator values.

All data analyses were performed using R version 3.3.2 (R Foundation for Statistical Computing, Vienna, Austria).

## Ethical considerations

As part of the original study, participants provided written informed consent. The study was reviewed and approved by the Mozambique National Bioethics Committee (338/CNBS/17) and the Office of the Associate Director for Science in the Center for Global Health at the Centers for Disease Control and Prevention (CGH2017-517).

## Results

Comparison of exit interview data with facility registry data showed poor overall sensitivity for key indicators, which in this study was defined as the number of indicators recorded in registries (for example presence of fever) divided by the number found during exit interviews (gold standard). Conversely specificity (true negatives observed during exit interviews recorded as such in the registries) was high for key indicators (**Table 1**). With the exception of microscopy results (which are deflated due to low numbers of facilities offering microscopy), the lowest sensitivity was for the recording of negative RDT results in Maputo, at 10%, while the highest sensitivity was seen for the recording of positive RDT results in Cabo Delgado, at 75%. Overall, sensitivities were substantially lower in Maputo than the other provinces, while similar values were seen in Zambézia and Cabo Delgado. Specificities were higher than sensitivities for all indicators, though the high values seen for microscopy indicators were affected by the low numbers of facilities offering microscopy. Cohen's Kappa was highest for the recording of positive RDT results in Cabo Delgado (0.67), and lowest for the recording of positive and negative microscopy results in Cabo Delgado and Zambézia, respectively (both at -0.01). As with the specificity measures, these values were affected by the low numbers of facilities offering microscopy.

Multivariable analysis of the factors associated with agreement between the re-examination and the facility registries on key indicators (**Table 2**) showed that patients with fever were more likely to be asked about fever during the initial consultation in the emergency department, compared with the triage ward (in Maputo and Cabo Delgado; adjusted Odds Ratio (aOR) = 0.75 and 0.39 (95% CI: 0.62–0.92, 0.22–0.68), respectively, for triage ward with emergency department as reference) and the outpatient ward (in Cabo Delgado, aOR = 0.37, 95% CI: 0.21–0.64). Fever results were also more likely to match in facilities with fewer than 1000 monthly consults in Cabo Delgado (aOR = 1.12, 95% CI: 1.05–1.18). Patients were more likely

**Table 1. Sensitivity, specificity, and Cohen's kappa of registry data as compared to gold-standard interviews with patients, Maputo, Zambézia, and Cabo Delgado Provinces, Mozambique, 2018.**

| | Maputo | | | Cabo Delgado | | | Cabo Delgado | | |
|---|---|---|---|---|---|---|---|---|---|
| Indicator | Sensitivity | Specificity | Cohen's Kappa | Sensitivity | Specificity | Cohen's Kappa | Sensitivity | Specificity | Cohen's Kappa |
| Presence of fever | 17 (13–21) | 95 (91–97) | 0.09 | 8 (5–10) | 95 (90–98) | 0.01 | 22 (18–26) | 86 (80–91) | 0.05 |
| RDT performed | 45 (36–54) | 94 (91–96) | 0.44 | 71 (67–76) | 81 (76–86) | 0.50 | 70 (66–75) | 70 (64–76) | 0.40 |
| RDT performed and positive | 58 (29–84) | 99 (97–100) | 0.55 | 75 (69–80) | 91 (88–93) | 0.67 | 70 (64–76) | 87 (84–90) | 0.58 |
| RDT performed and negative | 10 (5–17) | 98 (96–99) | 0.11 | 42 (35–50) | 94 (92–96) | 0.42 | 47 (39–55) | 88 (85–91) | 0.38 |
| Microscopy performed | 14 (4–36) | 100 (98–100) | 0.21 | 33 (11–65) | 99 (98–100) | 0.39 | 21 (6–51) | 99 (98–100) | 0.24 |
| Microscopy performed and positive | 0 (0–80) | 100 (99–100) | 0.00 | 0 (0–69) | 99 (98–100) | -0.01 | 17 (1–64) | 99 (98–100) | 0.15 |
| Microscopy performed and negative | 0 (0–20) | 100 (99–100) | 0.00 | 11 (1–49) | 100 (99–100) | 0.18 | 0 (0–40) | 100 (99–100) | -0.01 |

RDT: rapid diagnostic test

to have been administered an RDT (and have the results recorded) in all three provinces when patients had been managed according to national treatment guidelines during their initial exam with the health care provider (tested with RDT, given appropriate anti-malarials, aOR = 1.7, 1.6, and 1.8; 95% CI: 1.57–1.84, 1.47–1.74, 1.64–1.92; respectively in Maputo, Cabo Delgado, and Zambézia), and in facilities with fewer than 1000 monthly consults in Zambézia (aOR = 1.09, 95% CI: 1.00–1.18). Patients with positive RDT results were more likely to have these results match during re-examination for children under the age of 5 years in Zambézia and Cabo Delgado (aOR = 0.86 and 0.77, 95% CI: 0.79–0.94 and 0.71–0.84 for patients aged 5 years and older, with patients under 5 years as reference).

Retrospective data abstracted from health facility registers from November 2017 also showed discrepancies when compared with HMIS data from the same period (Table 3). The proportion of outpatient cases that were suspected and confirmed malaria were within 10% in registry and HMIS data in Maputo and Zambézia, though the difference between the proportion of outpatient cases with suspected malaria in registry and HMIS data in Cabo Delgado was 20%. In contrast, the absolute number of all-cause outpatient cases, suspected, and confirmed malaria cases were considerably different in all provinces; in all cases, numbers reported through the HMIS were higher than those abstracted from facility registries: across all three provinces, a total of 197,980 outpatient cases were abstracted from facility registries, compared with a total of 275,672 reported through the HMIS, for a difference of 77,692 cases.

**Table 2. Factors associated with quality of routine indicators recorded in health facility registers in Maputo, Cabo Delgado, and Zambézia provinces, Mozambique, 2018.**

|  |  | Maputo | | Cabo Delgado | | Zambézia | |
| --- | --- | --- | --- | --- | --- | --- | --- |
| **Fever Match (n = 1009)** |  | Adjusted Odds Ratio | 95% CI | Adjusted Odds Ratio | 95% CI | Adjusted Odds Ratio | 95% CI |
|  | Correctly managed | 1.08 | 0.94–1.24 | 1.04 | 0.98–1.10 | 1.08 | 0.99–1.19 |
|  | Facility under 1000 monthly consults | 0.98 | 0.84–1.14 | **1.12** | **1.05–1.18** | 1.03 | 0.93–1.13 |
|  | Patient aged 5 or older | 0.96 | 0.84–1.10 | 1 | 0.94–1.06 | 0.93 | 0.85–1.02 |
|  | Emergency department | Ref | Ref | Ref | Ref | Ref | Ref |
|  | Outpatient ward | - | - | **0.37** | **0.21–0.64** | 1.11 | 0.91–1.34 |
|  | Triage ward | **0.75** | **0.62–0.92** | **0.39** | **0.22–0.68** | 1.02 | 0.87–1.20 |
|  | Severe malaria case | 2.14 | 0.89–5.15 | 1.01 | 0.73–1.39 | - | - |
| **RDT Match (n = 1178)** |  |  |  |  |  |  |  |
|  | Correctly managed | **1.7** | **1.57–1.84** | **1.6** | **1.47–1.74** | **1.77** | **1.64–1.92** |
|  | Facility under 1000 monthly consults | 1.02 | 0.93–1.10 | 1.08 | 0.99–1.17 | **1.09** | **1.00–1.18** |
|  | Patient aged 5 or older | 1.01 | 0.94–1.09 | 0.99 | 0.91–1.08 | 0.93 | 0.86–1.01 |
|  | Emergency department | Ref | Ref | Ref | Ref | Ref | Ref |
|  | Outpatient ward | 1.15 | 0.64–2.07 | 0.82 | 0.35–1.94 | 0.9 | 0.76–1.06 |
|  | Triage ward | 1.03 | 0.58–1.84 | 0.82 | 0.35–1.94 | 0.97 | 0.84–1.11 |
|  | Severe malaria case | 0.98 | 0.65–1.48 | 0.97 | 0.59–1.60 | - | - |
| **Positive RDT result registered (n = 1325)** |  |  |  |  |  |  |  |
|  | Correctly managed | 1.01 | 0.97–1.06 | 1.04 | 0.96–1.14 | **1.14** | **1.05–1.24** |
|  | Facility under 1000 monthly consults | 1.01 | 0.96–1.06 | 1.02 | 0.93–1.12 | **1.16** | **1.06–1.26** |
|  | Patient aged 5 or older | 1.01 | 0.97–1.06 | **0.86** | **0.79–0.94** | **0.77** | **0.71–0.84** |
|  | Emergency department | Ref | Ref | Ref | Ref | Ref | Ref |
|  | Outpatient ward | 1.02 | 0.70–1.49 | 1.35 | 0.68–2.66 | 0.92 | 0.76–1.10 |
|  | Triage ward | 1.04 | 0.72–1.51 | 1.41 | 0.71–2.77 | 0.98 | 0.84–1.14 |
|  | Severe malaria case | 0.97 | 0.74–1.26 | 0.66 | 0.41–1.08 | - | - |

CI: Confidence intervals; RDT: Rapid diagnostic test; Ref: Reference

**Table 3. Comparison of key indicators between retrospective registry review and health management information system data for November 2017 in selected health facilities in Maputo, Zambézia, and Cabo Delgado Provinces, Mozambique.**

| Indicator | Maputo | | | Zambézia | | | Cabo Delgado | | |
|---|---|---|---|---|---|---|---|---|---|
| | Registry | HMIS | Difference | Registry | HMIS | Difference | Registry | HMIS | Difference |
| Proportion of patients with suspect malaria | 17% | 20% | 3% | 44% | 47% | 3% | 33% | 53% | 20% |
| Proportion of patients with malaria | 3% | 3% | 0% | 20% | 27% | 7% | 15% | 18% | 4% |
| Test positivity in suspect malaria cases | 19% | 14% | -5% | 46% | 58% | 12% | 44% | 34% | -10% |
| All-cause outpatient consults | 42,431 | 87,992 | 45,561 | 78,470 | 93,502 | 15,032 | 77,079 | 94,178 | 17,099 |
| Suspect malaria patients | 7,372 | 18,020 | 10,648 | 34,493 | 43,971 | 9,478 | 25,583 | 49,726 | 24,143 |
| Confirmed malaria cases | 1,370 | 2,435 | 1,065 | 15,876 | 25,298 | 9,422 | 11,266 | 17,127 | 5,861 |

HMIS: Health management information system

## Discussion

Health facility data on malaria cases are a key component of the NMCP's routine decision-making process; malaria case counts from facilities are routinely used for quantifying malaria commodity needs, creating risk maps showing national transmission patterns, guiding implementation of interventions, and measuring the impact of these interventions. The quality of this routine data can have significant impacts on the NMCP's ability to successfully combat malaria, though no recent studies have examined the quality of malaria data collected at health facilities. This study showed that the quality of malaria data collected at health facilities in Mozambique is not at the desired level, and that correlation between data recorded at facilities and data reported through the HMIS is less than ideal.

Comparisons of key indicators collected during re-examination and exit interviews with those collected from health facility registries showed little correlation. Although overall correlation was poor in all provinces, it was in general poorest in Maputo Province, mirroring the results from the quality of case management analysis [9]. The low sensitivity of fever data in all provinces is troubling, as it suggests that the clinicians are not asking about fever, not taking patients' temperature, or not recording these data, which means that the number of cases of malaria could be underestimated. The fact that positive RDT results were more likely to be recorded in facility registries than negative RDT results is not surprising, as similar trends have been seen with reporting of RDTs throughout Africa [11–13]. Notably, systematic under-reporting of negative RDTs can lead to overestimation of test positivity rate, which can bias quantifications of RDT and ACT needs. The fact that the sensitivity for positive RDT results in Maputo Province was only 58%, however, calls into question the estimated case counts for the area, as this is the key indicator used by the malaria program to estimate case totals. Furthermore, the fact that sensitivity of the indicators measured was consistently lower than specificity suggests that under-reporting, in particular of suspected and confirmed cases, is a bigger issue than over-reporting in the registry.

Mozambique has a standardized outpatient register where malaria testing is recorded but has experienced frequent register stock outs leading to the use of improvised documents that affects monthly data aggregation and register storage. At the time of data collection, many of the "records" that were included were sheets of loose-leaf paper or registers for other health areas that have been adjusted to be used for outpatients. Without a functional standardized system and materials for recording patient data, it is difficult, if not impossible, to know the exact number of expected registers or missing registers due to stockouts during a retrospective abstraction. Not being able to find all registers could lead to facilities not reporting all data on all malaria cases seen at the facility, resulting in under-estimated indicators in the HMIS on a

monthly basis and complicating subsequent DQA exercises. Mozambique has tried to address this challenge to DQA implementation by including a series of questions on register availability within its standardized DQA tool, an innovation that may be relevant for other settings in which data storage may be problematic. Although poor agreement was generally found between the gold standard re-examination/exit interview data and that recorded in facility registers, a number of indicators were more likely to be recorded properly in facility registries when patients were managed according to national treatment guidelines, suggesting that proper clinical training could influence data quality. This is supported by the fact that, as stated above, agreement between gold standard and facility registry data was poorest in Maputo Province, where adherence to treatment guidelines was also poorest [9]. Despite what has been seen in previous studies [14], the patient load of the facilities only influenced data quality in Zambézia. The fact that there were significant differences in the quality of data collected at different wards within facilities suggests there are instances where clinicians are paying more attention to the quality of the data they record than others or that there are cadres who record more accurately than others. Whether this is due to training and supervision visits such has been seen in community health worker DQAs [5,6], the presence of guidance materials, the size of the facility [15], or other factors could not be determined through this study. Further investigation of these findings could, however, provide guidance about how to improve data quality, for example, optimizing the type/frequency of in-service trainings, and ensuring proper monitoring and evaluation standard operating procedures are visible and available.

The inability to locate all registers or determine the number of missing registers during the retrospective data abstraction, which likely resulted in differences between abstracted values and values reported through the HMIS, also calls into question the reliability of how monthly summary data are tallied and reported. For all indicators, values abstracted from health facility records were lower than those reported through the HMIS. Although not being able to locate all registers clearly prevents the formation of strong conclusions based on these differences, when combined with the fact that percentage indicators (proportion of patients with suspected malaria, test positivity rate) were more similar than absolute value indicators (total number of suspected/confirmed patients), this suggests that registries at health facilities are being consistently misplaced, meaning that retrospective data quality assessments (DQA) would be consistently missing registers at the health facilities.

This study has a number of limitations. First, the data were collected as part of a study on case management practices, which prevented us from exploring the impact of health worker capacity on data quality. Second, while exit interviews minimize the risk of bias due to the Hawthorne effect, they are subject to bias due to patient recall; this could also have been an issue with respect to clinicians and their actions as well. [16]. As such, some of the findings of poor data quality could be due to patients' poor understanding of the survey questions or poor recall or understanding of what transpired during the patients' visit. Third, patient data were matched based on the name of the patient obtained during the exit interviews and those recorded in facility registries; because spelling of patient names was likely to be imperfect, this could have introduced some errors in the analysis. Finally, as mentioned above, data were abstracted from all registries that were found at the facilities, but in this study, it was not possible to quantify the expected (and therefore missing) number of registries at each facility. The study did not document whether there were stockouts of registers at the time of data collection and there is limited published evidence on this topic. Given the frequent stockouts of standardized registries noted anecdotally, missing data must be considered a distinct possibility, and a likely contributor to the consistently lower numbers found in registries compared with the HMIS.

These results suggest that care should be taken in interpreting trends based solely on routine data due to data quality issues. Nevertheless, simple steps such as providing consistent access to and storage of registers that include reporting of patient fever symptoms might improve the quality of routine data recorded at health facilities. As such, systematic, periodic health facility surveys, when done in conjunction with routine DQAs, can aid in evaluation of data quality and inform interpretation of routine data.

## Author Contributions

**Conceptualization:** James M. Colborn, Rose Zulliger, Frederico Brito, Gabriel Ponce de Leon, Abuchahama Saifodine, Baltazar Candrinho, Mateusz M. Plucinski.

**Data curation:** Marcel Andela.

**Formal analysis:** James M. Colborn, Mateusz M. Plucinski.

**Funding acquisition:** Frederico Brito.

**Investigation:** James M. Colborn, Rose Zulliger, Mariana Da Silva, Guidion Mathe, Ana Rita Chico, Frederico Brito, Marcel Andela, Gabriel Ponce de Leon, Baltazar Candrinho, Mateusz M. Plucinski.

**Methodology:** Rose Zulliger, Baltazar Candrinho, Mateusz M. Plucinski.

**Project administration:** Rose Zulliger, Mariana Da Silva, Guidion Mathe, Ana Rita Chico, Ana Christina Castel-Branco, Abuchahama Saifodine.

**Resources:** Rose Zulliger, Baltazar Candrinho.

**Supervision:** James M. Colborn, Rose Zulliger, Guidion Mathe, Ana Rita Chico, Ana Christina Castel-Branco, Frederico Brito, Gabriel Ponce de Leon, Abuchahama Saifodine, Baltazar Candrinho, Mateusz M. Plucinski.

**Writing – original draft:** James M. Colborn, Rose Zulliger, Baltazar Candrinho, Mateusz M. Plucinski.

**Writing – review & editing:** James M. Colborn, Rose Zulliger, Mariana Da Silva, Guidion Mathe, Ana Rita Chico, Ana Christina Castel-Branco, Frederico Brito, Marcel Andela, Gabriel Ponce de Leon, Abuchahama Saifodine, Baltazar Candrinho, Mateusz M. Plucinski.

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
