## [Decision Letter · Decision Letter 0]

3 Feb 2020

PONE-D-19-34921

Quality of Malaria Data in Public Health Facilities in Three Provinces of Mozambique

PLOS ONE

Dear Dr. Colborn,

Thank you for submitting your manuscript to PLOS ONE. After careful consideration, we feel that it has merit but does not fully meet PLOS ONE’s publication criteria as it currently stands. Therefore, we invite you to submit a revised version of the manuscript that addresses the points raised during the review process.

We would appreciate receiving your revised manuscript by Mar 19 2020 11:59PM. To enhance the reproducibility of your results, we recommend that if applicable you deposit your laboratory protocols in protocols.io, where a protocol can be assigned its own identifier (DOI) such that it can be cited independently in the future. For instructions see: http://journals.plos.org/plosone/s/submission-guidelines#loc-laboratory-protocols

We look forward to receiving your revised manuscript.

Kind regards,

Jennifer Yourkavitch

Academic Editor

PLOS ONE

Journal Requirements:

Additional Editor Comments (if provided):

In addition to responding to the reviewers' comments, please consider reading two papers on data quality published in the Journal of Global Health 2019 supplement on Rapid Access Expansion (RAcE) of integrated Community Case Management of malaria, pneumonia and diarrhea. The articles by Davis et al. and Yourkavitch et al. include information from data quality assessments in Mozambique and may provide relevant information to inform/reference in the background and conclusions of this paper.

Reviewers' comments:

Reviewer's Responses to Questions

**Comments to the Author**

1. Is the manuscript technically sound, and do the data support the conclusions?

Reviewer #1: Yes

Reviewer #2: Yes

2. Has the statistical analysis been performed appropriately and rigorously? 

Reviewer #1: Yes

Reviewer #2: Yes

3. Have the authors made all data underlying the findings in their manuscript fully available?

Reviewer #1: Yes

Reviewer #2: No

4. Is the manuscript presented in an intelligible fashion and written in standard English?

Reviewer #1: Yes

Reviewer #2: Yes

5. Review Comments to the Author

Reviewer #1: This is a well-written paper on an important topic that requires more attention by implementers and policy makers. The paper presents important information about malaria case data quality issues at the source of entry, an issue likely applicable more broadly to facility-level reporting. While this paper explicitly focuses on issues of recording patient-specific information into registries (Lines 91-94), it could be stronger by briefly addressing some of the potential system issues that could contribute to the reported results.

Consider adding a succinct definition of quality in the methodology section, particularly since the implicit definition used in this paper (high sensitivity and specificity and high Cohen’s Kappa) differs from that used in most DQAs.

In the methodology, please clarify the difference between laboratory and outpatient registers, and in which instances authors abstracted data from each if both were available at the facility. This is noted in the analysis section (different source form for different dichotomous outcomes). Assuming both outpatient and laboratory registers have report malaria test results, from which of these forms are data aggregated from individual facilities into the HMIS? Are there instances in which the authors found data recorded appropriately on one form and missing from the other? These system issues, if present, may contribute to the overreporting in the HMIS.

The results indicate significant overreporting of malaria cases in the HMIS, attributed primarily to issues of missing registers and poor recording by facility staff. Have the authors considered other contributing factors such as whether data from other sources (e.g. APE data for malaria (iCCM or CCMm)) are being reported into the HMIS at the district level, skipping facility-level aggregation? Did the authors find any missing data in the available registers? If so, this seems an important potential contributing factor. Missing data on testing and treatment may be a symptom of facility-level stockouts of RDTs and ACTs, which has been a widespread problem in Mozambique. I encourage the authors to consider addressing this in the results and conclusions, as appropriate.

Reviewer #2: Reviewer Recommendation and Comments for Manuscript Number PONE-D-19-34921

General Comment:

The authors present a methodologically sound analysis of the quality of both registry data as well as reporting of data from facility to HMIS in three provinces of Mozambique. Their overall conclusion about the weak quality of the malaria data is supported by the results presented and responds to a practical information gap. They have used methods previously reported on in the literature.

Details for consideration:

Overall, the writing is clear, however a few points have been identified that could benefit from further clarification.

1. Abstract

a. In second paragraph (lines 43-49) would be useful to mention the two lines of questioning the study undertakes: quality of registry data and quality of reporting

b. Line 54-55: note that gold standard has not been defined previously

c. Line 60-61: this does not align with relatively large difference between registry and HMIS suspected malaria indicator for Cabo Delgado (20%); for Maputo, the number 0.46% is not reported anywhere else in the manuscript—perhaps use this number in Table 3 for clarity/consistency?

d. Line 68-69: I think the authors could consider whether the statement about “access to registers” as a remedial measure is the most salient one to be drawn from this analysis (I did not see any data on the availability of registers, though there was mention in the discussion of the possibility of registers being misplaced—line 264—but this is not related to access to registers; the problem is cited a one of “data storage” so perhaps the recommendation should respond to this)

2. Introduction

a. Word count permitting, it would be helpful to incorporate a bit more background on state of the art practice in data quality assessment, and also why the approach to measuring quality was chosen

3. Methods

a. Line 113-114: to improve flow, introductory sentence could be modified to be more inclusive of the types of specific examples that follow (as is, it only refers to symptoms communicated, when the data actually collected goes beyond that)

b. Line 128: would help to include an explanation as to why the 6-month lag was selected (as opposed to a shorter lag)

c. Lines 148-155: why were these three outcomes selected to assess factors associated with matching data?

d. Lines 155-161: how/why were these predictor variables selected? None capture capacity/training of health worker.

4. Results

a. Line 203: for context could mention that November is high transmission period, as it is not mentioned in this paper

b. Table 3: was any information collected on registry stockouts that could provide context

5. Discussion

a. Line 236: Was it possible to determine whether register stock outs occurred during the period of the study?

b. Line 244-245: interpretation appears to be overstated as “correctly managed” factor was only statistically significant for one indicator (‘RDT match’) across provinces (and in one province for ‘RDT result registered’)

c. Line 249-252: the differences in quality of data collected at different wards was only statistically significant for one of the outcomes tested (fever match)

d. Line 277: The authors could reconsider whether the statement about “access to registers” as a remedial measure is the most salient one to be drawn from this analysis. They did not report any data on the availability/access to registers, though there was mention in the discussion of the possibility of registers being misplaced—line 264; the problem is cited a one of “data storage” and record keeping/management so perhaps a recommendation should respond to this.

e. Lines 252-254: Inability to explore influence of health worker capacity (e.g. data management and service provision) on data quality could be considered a limitation if this is due to the fact that data was not collected on such topics

f. Line 267: interestingly, the source cited (16) notes that problems with exit interviews resulting in recall bias occur primarily with respect to collecting information about “provision of clinical advice”; this limitation was not reported for information relating to “provision of clinical actions” (such as taking an RDT test, prescribing medication)

g. Generally, the reader is not left with a clear message about what should be done with these conclusions—practical implications for improving data quality in Mozambique (only mention improving access to registers)

h. It would be useful to see something about future research that could complement this study to understand the “why” behind the data quality issues identified in this paper.

Compliance with PLOS One Data Policy:

It appears that there are some restrictions to the data used in this study.

6. PLOS authors have the option to publish the peer review history of their article (what does this mean?). If published, this will include your full peer review and any attached files.

Reviewer #1: No

Reviewer #2: No

---

## [Author Response · Author response to Decision Letter 0]

19 Mar 2020

Reviewer #1

Comment:

“While this paper explicitly focuses on issues of recording patient-specific information into registries (Lines 91-94), it could be stronger by briefly addressing some of the potential system issues that could contribute to the reported results.”

Author’s response:

We have tried to revise the language of the manuscript in order to highlight some of the system issues that could contribute to the reported results.

Comment:

“Consider adding a succinct definition of quality in the methodology section, particularly since the implicit definition used in this paper (high sensitivity and specificity and high Cohen’s Kappa) differs from that used in most DQAs”

Author’s response:

Done.

Comment:

“In the methodology, please clarify the difference between laboratory and outpatient registers, and in which instances authors abstracted data from each if both were available at the facility. This is noted in the analysis section (different source form for different dichotomous outcomes). Assuming both outpatient and laboratory registers have report malaria test results, from which of these forms are data aggregated from individual facilities into the HMIS? Are there instances in which the authors found data recorded appropriately on one form and missing from the other? These system issues, if present, may contribute to the overreporting in the HMIS.”

Author’s response: 

As suggested, the difference between laboratory and outpatient registers has been clarified in the methods section.

Outpatient registers record actual case information (patients with fever, patients tested, patients positive, etc), while laboratory registers record test results only. All case data in HMIS comes from facility registers, while test results (total tests used, total positive) in HMIS come from laboratory registers. Therefore actual case data does not come from both registers, though testing results in HMIS from outpatient registers and laboratory registers rarely align, thus further validating the issues with data recording and quality.

Comment:

“The results indicate significant overreporting of malaria cases in the HMIS, attributed primarily to issues of missing registers and poor recording by facility staff. Have the authors considered other contributing factors such as whether data from other sources (e.g. APE data for malaria (iCCM or CCMm)) are being reported into the HMIS at the district level, skipping facility-level aggregation? Did the authors find any missing data in the available registers? If so, this seems an important potential contributing factor. Missing data on testing and treatment may be a symptom of facility-level stockouts of RDTs and ACTs, which has been a widespread problem in Mozambique. I encourage the authors to consider addressing this in the results and conclusions, as appropriate.”

Author’s response:

The reviewer makes a good point that if APE data is being entered directly into the HMIS (or entered at the district level) and skipping facility registers, then this could contribute to the overreporting seen in the HMIS. However, this was not an issue in this analysis as APE data are separated from health facility data in HMIS in Mozambique. As such, health facility data was abstracted from only outpatient registers, and was compared to only outpatient data from the HMIS (without APE data).

Reviewer #2: Reviewer Recommendation and Comments for Manuscript Number PONE-D-19-34921

General Comment:

The authors present a methodologically sound analysis of the quality of both registry data as well as reporting of data from facility to HMIS in three provinces of Mozambique. Their overall conclusion about the weak quality of the malaria data is supported by the results presented and responds to a practical information gap. They have used methods previously reported on in the literature.

Details for consideration:

Overall, the writing is clear, however a few points have been identified that could benefit from further clarification.

1. Abstract

a. In second paragraph (lines 43-49) would be useful to mention the two lines of questioning the study undertakes: quality of registry data and quality of reporting

b. Line 54-55: note that gold standard has not been defined previously

c. Line 60-61: this does not align with relatively large difference between registry and HMIS suspected malaria indicator for Cabo Delgado (20%); for Maputo, the number 0.46% is not reported anywhere else in the manuscript—perhaps use this number in Table 3 for clarity/consistency?

Author’s response:

We have made the changes suggested in a and b. The 0.46% stated in the abstract was an error, and has been fixed. While we agree that 20% is much larger than the 3% difference seen in the other provinces, we still consider it relatively small compared with the differences seen in the other indicators. Nonetheless we have adjusted the language in the abstract.

d. Line 68-69: I think the authors could consider whether the statement about “access to registers” as a remedial measure is the most salient one to be drawn from this analysis (I did not see any data on the availability of registers, though there was mention in the discussion of the possibility of registers being misplaced—line 264—but this is not related to access to registers; the problem is cited a one of “data storage” so perhaps the recommendation should respond to this)

Author’s response:

We have revised the language to more closely relate to the results and the content within the discussion section. We have also referenced the potential role that storage might have on data quality, as documented during subsequent data quality assessments.

2. Introduction

a. Word count permitting, it would be helpful to incorporate a bit more background on state of the art practice in data quality assessment, and also why the approach to measuring quality was chosen

Author’s response:

We have added information on the introduction explaining why data quality assessments are important with specific information on the approach deployed in the present study.

3. Methods

a. Line 113-114: to improve flow, introductory sentence could be modified to be more inclusive of the types of specific examples that follow (as is, it only refers to symptoms communicated, when the data actually collected goes beyond that)

b. Line 128: would help to include an explanation as to why the 6-month lag was selected (as opposed to a shorter lag)

c. Lines 148-155: why were these three outcomes selected to assess factors associated with matching data?

Author’s response:

We have made the suggested changes noted in a and b. Regarding c, these were the three outcomes that had the highest numbers of entries, and therefore presented the best chance to find significant associations with the predictor variables. Language has been added to the manuscript to make this more clear.

d. Lines 155-161: how/why were these predictor variables selected? None capture capacity/training of health worker.

Author’s response:

Because this study was originally designed to measure case management practices, it unfortunately did specifically not collect data on health worker training or capacity. The variables selected were those that were available either through the study itself or other data sources (such as the HMIS).

4. Results

a. Line 203: for context could mention that November is high transmission period, as it is not mentioned in this paper

Author’s response:

November is technically the end of the low transmission season, with cases usually beginning to increase in December or January. Text clarifying this has been added to the manuscript.

b. Table 3: was any information collected on registry stockouts that could provide context

Author’s response:

Unfortunately not. We have noted this as a limitation in the discussion section.

5. Discussion

a. Line 236: Was it possible to determine whether register stock outs occurred during the period of the study?

Author’s response:

Unfortunately not. We have noted this as a limitation in the discussion section.

b. Line 244-245: interpretation appears to be overstated as “correctly managed” factor was only statistically significant for one indicator (‘RDT match’) across provinces (and in one province for ‘RDT result registered’)

Author’s response:

We have added more nuance to our interpretation of this indicator. 

c. Line 249-252: the differences in quality of data collected at different wards was only statistically significant for one of the outcomes tested (fever match)

Author’s response:

This is a good point, though given that testing for fever is a key component of the case management pathway, we feel that describing these results in this manner and highlighting potential causes for not testing for fever is important.

d. Line 277: The authors could reconsider whether the statement about “access to registers” as a remedial measure is the most salient one to be drawn from this analysis. They did not report any data on the availability/access to registers, though there was mention in the discussion of the possibility of registers being misplaced—line 264; the problem is cited a one of “data storage” and record keeping/management so perhaps a recommendation should respond to this.

Author’s response:

We agree that registers being misplaced could be a potential problem, though without standardized registers in use at facilities it is difficult if not impossible to identify/quantify missing registers. As a consequence we have included the use of standardized registers as a primary recommendation stemming from this work, which while it may not directly address the problem of missing registers will at least enable the problem to be quantified. Nevertheless, we have also emphasized the important role that storage might have had in data availability.

e. Lines 252-254: Inability to explore influence of health worker capacity (e.g. data management and service provision) on data quality could be considered a limitation if this is due to the fact that data was not collected on such topics

Author’s response:

This is a valid point and language stating this has been added to the manuscript.

f. Line 267: interestingly, the source cited (16) notes that problems with exit interviews resulting in recall bias occur primarily with respect to collecting information about “provision of clinical advice”; this limitation was not reported for information relating to “provision of clinical actions” (such as taking an RDT test, prescribing medication)

This is also a good point that has been added to the manuscript.

g. Generally, the reader is not left with a clear message about what should be done with these conclusions—practical implications for improving data quality in Mozambique (only mention improving access to registers)

h. It would be useful to see something about future research that could complement this study to understand the “why” behind the data quality issues identified in this paper.

Author’s response:

Language to address both points above have been added to the manuscript.

---

## [Editor Report · Decision Letter 1]

23 Mar 2020

Quality of Malaria Data in Public Health Facilities in Three Provinces of Mozambique

PONE-D-19-34921R1

Dear Dr. Colborn,

We are pleased to inform you that your manuscript has been judged scientifically suitable for publication and will be formally accepted for publication once it complies with all outstanding technical requirements.

With kind regards,

Jennifer Yourkavitch

Academic Editor

PLOS ONE
---

## [Editor Report · Acceptance letter]

7 Apr 2020

PONE-D-19-34921R1 

Quality of Malaria Data in Public Health Facilities in Three Provinces of Mozambique 

Dear Dr. Colborn:

I am pleased to inform you that your manuscript has been deemed suitable for publication in PLOS ONE. Congratulations! Your manuscript is now with our production department. 

With kind regards,

on behalf of

Dr. Jennifer Yourkavitch 

Academic Editor

PLOS ONE